# Advancements in Prenatal Genetic Screening and Testing: Emerging Technologies and Evolving Applications

**DOI:** 10.3390/diagnostics15202579

**Published:** 2025-10-13

**Authors:** Mona M. Makhamreh, Mei Ling Chong, Ignatia B. Van den Veyver

**Affiliations:** 1Department of Obstetrics and Gynecology, Baylor College of Medicine, Houston, TX 77030, USA; 2Department of Molecular and Human Genetics, Baylor College of Medicine, Houston, TX 77030, USA; 3Duncan Neurological Research Institute at Texas Children’s Hospital, 1250 Moursund Street, Room 1025.14, Houston, TX 77030, USA

**Keywords:** prenatal genetic testing, sequencing, cfDNA, genomics

## Abstract

Advancements in genomic technologies have transformed prenatal genetic testing, offering more accurate, comprehensive, and noninvasive approaches to reproductive care. This review provides an in-depth overview of current methodologies and emerging innovations, including expanded carrier screening (ECS), cell-free DNA (cfDNA) testing, chromosomal microarray analysis (CMA), and sequencing-based diagnostics. We highlight how next-generation sequencing (NGS) technologies have revolutionized carrier screening and fetal genome analysis, enabling detection of a broad spectrum of genetic conditions. The clinical implementation of cfDNA has expanded from common aneuploidies to include copy number variants (CNVs), and single-gene disorders. Diagnostic testing has similarly evolved, with genome sequencing outperforming traditional CMA and exome sequencing through its ability to detect both sequence and structural variants in a single assay. Emerging tools such as optical genome mapping, RNA sequencing, and long-read sequencing further enhance diagnostic yield and variant interpretation. This review summarizes major technological advancements, assesses their clinical utility and limitations, and outlines future directions in prenatal genomics.

## 1. Introduction

Over the last decade, rapid advancements in prenatal genetics have transformed the landscape of reproductive care, largely driven by high-throughput next-generation sequencing (NGS). This technology has revolutionized genomic screening and diagnostics by enabling parallel sequencing of multiple genomic regions [1]. Several key assays now form the foundation of modern prenatal screening and diagnosis (Figure 1). Expanded carrier screening (ECS) enables pan-ethnic detection of a broad range of autosomal recessive and X-linked conditions in prospective parents and is performed on a maternal blood sample [2]. Cell-free DNA (cfDNA) screening, also based on a maternal blood sample, noninvasively assesses fetal risk for common aneuploidies, and with advancing technology has expanded to include sex chromosome aneuploidy (SCA) [3], rare autosomal trisomies (RATs) [4], both common and rare copy number variants (CNVs) [5], and single-gene disorders [6]. Among diagnostic tests performed prenatally, usually done on samples obtained through chorionic villus sampling (CVS) or amniocentesis, chromosomal microarray analysis (CMA), which detects aneuploidy and CNVs, including submicroscopic CNVs not visible through conventional karyotyping, is the recommended diagnostic test for pregnancies complicated by fetal anomalies [7]. More recently, low-pass genome sequencing using NGS has been shown to detect CNVs with broader coverage compared to traditional CMA in the prenatal setting [8]. Additionally, NGS-based tests can target a selected panel of genes, the exome, or the entire genome at nucleotide-level resolution, enabling the detection of a wide array of genetic conditions during pregnancy [9]. Methylation assays are also clinically available for prenatal diagnosis of imprinting disorders, though a detailed discussion is beyond the scope of this review [10]. Genetic variants are classified as pathogenic, likely pathogenic, of uncertain significance, likely benign, or benign [11]. Based on American College of Medical Genetics and Genomics/Association for Molecular Pathology (ACMG/AMP) guidelines, laboratories are recommended to report only pathogenic and likely pathogenic variants; however, variants of uncertain significance (VUS) may be reported in certain clinical contexts [11]. In parallel with these technologies, emerging methods such as optical genome mapping, RNA sequencing, and long-read sequencing are beginning to expand diagnostic possibilities and—along with advancements in bioinformatics—can enhance variant interpretation. These advancing tools, while not yet routine, mark the next frontier in prenatal genomics. This review outlines the evolution of key assays, recent technological advances, and their clinical utility and limitations in prenatal screening and diagnosis.

This review was informed by a targeted literature search of PubMed and Google Scholar using key terms such as “prenatal genetics,” “carrier screening,” “cell-free DNA,” “chromosomal microarray,” and “exome sequencing.” Priority was given to publications from the last five years, supplemented by landmark earlier studies and current clinical guidelines. Articles were selected for relevance to prenatal applications, clinical utility, and technological advances. The review was not intended as a systematic review, but rather as an overview of key developments and evolving applications in the field.

## 2. Expanded Carrier Screening

### 2.1. Overview of Carrier Screening

Carrier screening identifies usually asymptomatic individuals and their partners who carry pathogenic variants for autosomal recessive or X-linked conditions. While ideally performed preconceptionally, it is often offered in early pregnancy as part of routine obstetric care to assess the risk of having an affected offspring [12,13]. Historically, carrier screening was limited to conditions with high prevalence in specific populations, such as Tay–Sachs disease in Ashkenazi Jewish communities and β-thalassemia in Mediterranean populations [13]. The discovery of the *CFTR* gene prompted widespread cystic fibrosis screening, now recommended pan-ethnically. Spinal muscular atrophy (SMA) screening followed a similar path [14]. Traditional criteria for screening included severe phenotype, high carrier frequency, reliable testing, clear genotype–phenotype correlation, and access to prenatal diagnosis and reproductive choices [2,12].

### 2.2. Technological Evolution, Challenges, and Limitations

How we approach genetic screening has been reshaped by the rapidly advancing technology used for DNA sequencing, allowing for high throughput with rapid turnaround times and lower costs of genome sequencing [15]. Traditional carrier screening relied on simpler methods, including polymerase chain reaction-based techniques, multiplex ligation-dependent probe amplification, microarrays, Sanger sequencing, and genotyping. NGS now enables simultaneous screening of large numbers of genes in carrier screening panels covering tens to hundreds of genes. In practice, panels often incorporate additional methods, such as copy number analysis for genes where copy number changes are a significant cause of disease, with assay selection tailored to each gene or variant based on feasibility, reliability, and cost [16].

Although NGS enables carrier screening for many recessive or X-linked diseases simultaneously, developing a multigene panel that meets carrier screening criteria along with known positive and negative predictive values for each can be challenging [17]. Each test should be thoroughly validated to define its analytical sensitivity, specificity, and accuracy, and establish confidence in the detection, analysis, and reporting of genetic variants [18]. Additionally, certain clinically significant genes present technical challenges for NGS analysis due to the presence of pseudogenes (for example, *GBA* for Gaucher disease), repeat expansions (*FMR1* for fragile X syndrome), or DNA structural variations [19].

### 2.3. Guidelines and Ethical Considerations

The American College of Obstetricians and Gynecologists (ACOG) recommends universal screening for cystic fibrosis and SMA and for hemoglobinopathies via complete blood count, with additional screening based on ethnicity or family history (e.g., Tay–Sachs disease and fragile X syndrome). ECS for conditions with a population carrier frequency of ≥1/100 is considered acceptable, though providers are advised to standardize their approach for each patient [12]. In contrast, the ACMG recommends pan-ethnic carrier screening to promote equity and proposed a tier-based classification for carrier screening [2]. Tier 1 aligns with the ACOG’s recommendations, tier 2 includes conditions with a carrier frequency ≥1/100, tier 3 includes conditions with a frequency ≥1/200 (including some X-linked conditions), and tier 4 covers rare conditions without frequency thresholds [12,20,21]. The ACMG recommends tier 3 screening for all individuals and tier 4 for those with higher-risk factors such as family history or consanguinity [2]. Despite these guidelines, ECS panel content varies across laboratories [22]. Increasing multiethnicity and population admixture, inaccuracies in self-reported ethnicity, and growing emphasis on patient autonomy all support using larger pan-ethnic ECS panels [13,23,24,25]. However, larger panels may yield uncertain results, increase parental anxiety, and lead to additional testing and costs [26]. Furthermore, accurate residual risk counseling remains challenging due to limited data on disease-specific carrier frequencies across diverse populations. These factors highlight the importance of clear, well-documented counseling and informed consent before carrier screening [27].

### 2.4. Carrier Screening by Genome Sequencing

Data on genome sequencing for carrier screening are still limited, but improved detection rates for certain conditions compared to traditional panels have been reported [28]. In one study 202 individuals underwent preconception carrier screening using genome sequencing, but result reporting was limited to a curated list of 728 single-gene disorders. Although the test relied on genome sequencing, reporting was constrained to the predefined panel, limiting its scope and making it not a true application of comprehensive genome sequencing. Nonetheless, this approach identified 304 clinically significant variants, demonstrating improved sensitivity over traditional panels [29]. A separate study used duo-exome sequencing of parents after fetal or embryonic loss to identify shared carrier status for pathogenic variants, offering “molecular autopsy by proxy” when fetal DNA was unavailable. This approach enabled diagnosis of autosomal recessive conditions underlying the pregnancy loss and mimics the concept of genome-wide carrier screening. While both studies highlight potential applications of broad sequencing, the clinical utility of genome sequencing for routine carrier screening remains to be fully explored [30].

## 3. Cell-Free DNA

### 3.1. Overview of cfDNA Screening for Aneuploidies and Copy Number Variants

Prenatal screening by cfDNA analysis for the detection of aneuploidies became clinically available in 2011 and has been rapidly introduced into prenatal care to detect common aneuploidies (chromosomes 13, 18, and 21) [31,32,33]. Prior to implementation of cfDNA-based aneuploidy screening, biochemical serum screening (β-hCG, AFP, PAPP-A, uE3, inhibin A), often combined with nuchal translucency, formed the basis of combined, triple, and quadruple screens for aneuploidy risk, particularly trisomies 21 and 18, and remains in use in some settings [33]. Circulating fetal cfDNA in maternal plasma was initially described in 1997 [34], and was shown to be trophoblast-derived. The fetal fraction describes the proportion of all cfDNA circulating in maternal blood that is of placental (trophoblast) origin, comprises about 3%–13% of total cfDNA, and increases throughout gestation [35,36]. Multiple studies have consistently shown the superior performance of prenatal cfDNA screening for common aneuploidies compared to traditional maternal serum screening, and it has been endorsed by professional societies for that use [3,5,33].

cfDNA screening for aneuploidy uses two NGS-based methods. In the first, massively parallel sequencing randomly sequences cfDNA from maternal plasma, map reads to the reference genome, and uses counting statistics to detect chromosomal imbalances [31]. The second, targeted sequencing, focuses on single-nucleotide polymorphism (SNP)-rich regions using capture probes or multiplex PCR, with statistical analysis of read counts and allele distributions to identify chromosomal abnormalities [37,38]. With advancements in cfDNA screening, the test has also been validated for use in twin pregnancies, with detection rates for trisomy 21 approaching those seen in singletons, while test performance for trisomies 18 and 13 is slightly lower. cfDNA screening is also available for higher-order multiples [5,39]. Additionally, analysis of SNPs and fetal-specific alleles can determine zygosity in twin pregnancies [40]. Circulating cfDNA from a vanishing twin can persist for weeks and confound screening results due to the persistence of cfDNA from a demised fetus [41].

Noninvasive fetal CNV screening from cfDNA in maternal blood typically follows one of two approaches: targeted analysis of known microdeletion/duplication syndromes or broader genome-wide screening. In the United States, genome-wide CNV screening offered by one commercial provider is designed to detect CNVs larger than 7 megabases (Mb) and includes selected known clinically relevant microdeletions smaller than 7 Mb [5], but many other pathogenic CNVs also fall below this size threshold [7]. Test performance of cfDNA screening depends on several factors, including the prevalence of chromosomal abnormality, presence of maternal CNVs, placental mosaicism, fetal fraction, sequencing depth, and CNV size [4,5,42]. Most individual CNVs are rare, resulting in lower sensitivity and specificity of cfDNA screening compared to common trisomies and SCAs [3]. Among pathogenic CNVs, 22q11.2 deletion syndrome (22q11.2DS) is the most common one identified prenatally [43], resulting in better performance metrics for this condition. A recent study of 18,289 pregnancies assessed SNP-based cfDNA screening for 22q11.2DS and detected 10 of 12 confirmed cases using an updated algorithm. With a 1-in-100 risk cutoff, the false-positive rate was 0.05% and the positive predictive value (PPV) 52.6% [44]. Currently, there is insufficient evidence to support routine screening for CNVs other than 22q11.2DS [5].

Genome-wide CNV screening with cfDNA analysis can detect RATs that are trisomies involving chromosomes other than 13, 18, 21, X, or Y. Liveborn infants with non-mosaic RATs are extremely rare, and when detected prenatally, RATs are most often mosaic [5]. A recent meta-analysis of 31 studies reviewed results from 1703 high-risk cfDNA screening results for RATs and reported a pooled PPV of 11.46% [45]. Recent studies have shown that RATs, when present as confined placental mosaicism (CPM), are associated with adverse pregnancy outcomes, including fetal growth restriction, low birth weight, and preeclampsia, emphasizing the need for optimal antenatal surveillance and counseling in pregnancies when RATs are detected by cfDNA testing [46,47]. Despite these technical capabilities, evidence regarding the clinical validity of genome-wide CNV or aneuploidy screening by cfDNA remains limited [46], and its use in pregnant individuals has been the subject of scientific debate [48]. It is currently not recommended by professional societies in the U.S. [5,33]. cfDNA screening has expanded beyond common autosomal trisomies to routinely include SCAs. Given the unique clinical complexities of SCAs, including variable phenotypic expression, mosaicism, and lower PPV, several reports have underscored the importance of thorough pre- and post-test counseling and need for confirmatory testing [5,49].

### 3.2. Maternal cfDNA Testing and Screening for Fetal Single Gene Disorders

Single-gene disorders (SGDs) affect approximately 1% of births [50]. cfDNA testing has been developed for the noninvasive prenatal diagnosis of SGD, most commonly through targeted assays for known familial or suspected variants, encompassing both autosomal dominant and recessive conditions. Additionally, cfDNA-based screening panels have been primarily designed to detect dominant, often de novo, pathogenic variants associated with severe, early-onset phenotypes [51]. More recently, genome-wide noninvasive fetal sequencing approaches have been explored for the comprehensive detection of monogenic disorders [52]. It is important to distinguish between: (1) noninvasive prenatal diagnosis (NIPD) by cfDNA analysis for SGD diagnosis, targeted for known variants; and (2) cfDNA analysis for SGD screening, primarily for dominant conditions [51,52].

In one study that evaluated 422 cases using the cfDNA-SGD assay, which targets 30 genes associated with dominant SGDs, there were 20 true positives and 127 true negatives among 147 cases with available confirmatory results. Notably, no false-positive or false-negative results were identified in this cohort [53]. In a separate study of 2208 individuals with high-risk pregnancies, from cfDNA-based SGD testing with the same panel of 30 genes, 125 (5.7%) tested positive. The highest detection rates were observed for pregnancies referred for fetal long-bone abnormalities (33.7%), craniofacial malformations (28.6%), and a family history of a disorder included in the panel (15.2%). Among those with confirmatory diagnostic testing, no false-positive or false-negative results were reported [6]. In a study of 2745 individuals with low-risk pregnancies screened with this cfDNA-based SGD assay, 14 (0.51%) had high-risk results. While no false positives were observed, there were discrepancies in variant classification between cfDNA results and diagnostic testing for two of them. The study highlighted both the potential for early detection and the counseling challenges posed by variable expressivity, uncertain genotype–phenotype correlations, and interpretive differences [54]. While these findings are promising, broader adoption will require population-based validation, equitable access, and critical assessment of ethical, economic, and policy-related challenges. A comprehensive framework encompassing rigorous test design, standardized counseling, and informed consent is essential to support responsible implementation and improve clinical outcomes [51].

### 3.3. Developments in cfDNA Analysis

cfDNA tests for noninvasive fetal rhesus (Rh) genotyping have been introduced into obstetric care in different countries. In the U.S., one multicenter cohort study evaluated the accuracy of cfDNA-based fetal red blood cell antigen genotyping in 156 alloimmunized pregnancies. Testing between 10 and 37 weeks’ gestation showed complete concordance with postnatal neonatal genotyping across 465 antigen calls, yielding 100% sensitivity, specificity, and accuracy. These findings support the use of cfDNA testing from 10 weeks’ gestation to guide management and reduce unnecessary interventions [55]. The evidence base for cfDNA-based Rh testing is extensive and has a long-standing track record in European clinical practice prior to its introduction in the U.S. Broader adoption also enables targeted Rh immunoglobulin administration by identifying RhD-negative pregnancies carrying RhD-positive fetuses [56].

In a recent report, one U.S. commercial laboratory developed and evaluated the effectiveness of a maternal carrier screening test for cystic fibrosis, hemoglobinopathies, and spinal muscular atrophy followed by cfDNA-based single-gene testing when the mother is identified as a carrier for one of these conditions [57]. This strategy can provide a fetal risk assessment for these recessive conditions without the requirement for paternal carrier screening. This facilitates prenatal counseling and management while addressing the logistical challenges of obtaining paternal carrier status [57]. It reported a sensitivity of 96%, specificity of 95.2%, positive predictive value of 50.0%, and negative predictive value of 99.8% in a study of 42,067 individuals [58]. However, the number of genes included in this carrier screening with the cfDNA analysis approach is well below the ACMG recommended tier 3 panel.

Fragmentomics refers to the analysis of cfDNA fragmentation patterns. Fetal (placental) cfDNA fragments are typically shorter than maternal cfDNA, and this size difference has been leveraged to estimate fetal fraction and detect fetal aneuploidies [59,60]. In cases of fetal trisomy, the affected chromosome contributes to a shift in the fragment length distribution depending on whether there is a chromosomal gain or loss. Integrating size-based and count-based analytical approaches improves the precision of cfDNA screening by aiding in the interpretation of CNVs [61]. Current NGS-based cfDNA screening typically requires a fetal fraction greater than 4%. Fetal fraction amplification, which enriches placental cfDNA in maternal plasma based on fragment size, can help overcome maternal DNA background interference [62]. This technique may be especially useful in cases where the fetal fraction is reduced, such as with a high maternal body mass index, or at earlier gestational ages. Studies have demonstrated that fetal fraction amplification improves the analytical performance of cfDNA, reduces test failures due to low fetal fraction, enables higher-resolution detection of fetal chromosomal abnormalities and CNVs, and enables cfDNA screening at an earlier gestational age [63,64].

A newer promising development is genome-wide noninvasive fetal sequencing (NIFS) on maternal plasma cfDNA [51,52]. Two proof-of-concept studies demonstrated the feasibility of exome-based noninvasive prenatal testing using cfDNA for detecting single-nucleotide variants (SNVs), indels, and select CNVs. Sensitivity was highest for de novo and paternally inherited variants, with reduced accuracy at low fetal fractions and for maternally inherited alleles. Trio-based sequencing achieved 100% concordance with invasive testing in high-risk pregnancies. These findings support the potential future expansion of NIFS to detect monogenic disorders, but also underscore the need for confirmatory testing of any detected variants, larger validation studies, and technical and cost improvements [65,66].

There are instances when cfDNA screening can yield unexpected results. Retrospective studies from large commercial and national laboratories have associated unusual sequencing patterns, such as multiple aneuploidies or autosomal monosomies, with underlying maternal malignancy [67,68]. When a tumor is present, it may shed cfDNA into maternal plasma, contributing to the total cfDNA pool and distorting expected chromosomal ratios, thereby complicating interpretation. Such discordant results often lead to a nonreportable outcome, as fetal aneuploidy status cannot be confidently determined. A recent study of 107 pregnant individuals with unusual or nonreportable cfDNA results found that 52 (48.6%) were subsequently diagnosed with occult cancer. Among 49 participants whose sequencing showed copy number alterations across three or more chromosomes, 47 (95.9%) had a confirmed malignancy, which was best detected by whole-body MRI [69]. As cfDNA screening expands in scope, understanding its limitations and incidental findings becomes increasingly important.

## 4. Chromosomal Microarray and Copy Number Variant Detection

CMA is the recommended test for patients undergoing prenatal diagnostic testing when one or more major fetal structural abnormalities are identified by ultrasonography or pregnancies with an intrauterine fetal demise or stillbirth [70]. CMA detects clinically significant deletions and duplications, can reveal regions of homozygosity (ROH), which represent copy-neutral loss of heterozygosity (CN-LOH), and may suggest uniparental disomy or parental consanguinity.

Among samples with a nondiagnostic karyotype, CMA identified clinically significant deletions or duplications in 6.0% of fetuses with structural anomalies with higher yields when there were multiple anomalies and in 1.7% of fetuses where testing was done for advanced maternal age or positive screening results [7,71]. The interpretation of VUS remains a challenge, potentially complicating genetic counseling and decision-making during pregnancy. CMA has additional limitations: it cannot detect balanced chromosomal rearrangements, such as balanced translocations or inversions, has limited sensitivity for low-level mosaicism (<10%–20%), and cannot reliably detect triploidy unless SNP probes are included [70]. Furthermore, CMA cannot reveal the underlying mechanism of a chromosomal imbalance, such as trisomies. These still require conventional karyotyping to determine if the imbalance resulted from meiotic non-disjunction or an imbalanced Robertsonian translocation, which is critical for assessing recurrence risk in future pregnancies [7,70]. CMA does not detect SNVs, small indels, or small CNVs below platform resolution [70] either. The general consensus for the reportable size threshold in prenatal CMA is 200 to 400 Kb, with some suggesting even more conservative cutoffs (e.g., 500 Kb for deletions and 1 Mb for duplications) to reduce the detection of VUS that could result in associated parental anxiety [72]. A 2023 study using high-resolution CMA platforms found that reducing the size threshold to 20 Kb can uncover pathogenic CNVs below the standard reporting cutoff while keeping VUS low at 4%, supporting the view that higher-resolution platforms can improve diagnostic yield without significantly increasing uncertainty [73]. Given the rapidly increasing number of gene–disease associations and continuous updates in genetic databases, adopting higher-resolution CMA testing in prenatal diagnostics is essential for accurately identifying gene-level CNVs.

Although CMA is widely used for prenatal CNV detection, it is limited by a preset probe design and a workflow that does not readily incorporate new gene–disease associations. As more laboratories adopt NGS, low-pass genome sequencing (LP-GS) has emerged as a promising alternative, offering flexible CNV detection with reported coverage in prenatal studies ranging from 0.25× to 5× [8,74,75,76]. Prenatal studies comparing LP-GS with CMA have reported diagnostic yields ranging from 8% to 23% [8,76,77]. Although the fundamental principles and clinical utility of LP-GS have been established, broader adoption may require further validation and endorsement by professional guidelines. In the long run, LP-GS holds potential as a more cost-effective and efficient diagnostic tool, particularly when integrated into a comprehensive sequencing strategy.

## 5. Diagnostic Sequencing

Since the advent of NGS, prenatal genetic diagnostics have undergone significant transformation. NGS-based tests can target a selected panel of genes, the exome, or the entire genome at a nucleotide level, enabling the detection of a wide array of genetic conditions during pregnancy.

### 5.1. Targeted Gene Panels

Targeted gene panels focus on sequencing specific sets of genes associated with particular phenotypes or conditions, such as a skeletal dysplasia [78] and RASopathies [79]. However, their diagnostic utility can be limited by gene content. In a cohort of 127 NIHF cases, exome sequencing (ES) identified pathogenic variants in 29% of fetuses, while targeted panels would have captured only 11% to 62% of these variants depending on the panel composition [80]. In addition, selecting an appropriate panel testing strategy heavily depends on ultrasound findings and the suspected diagnosis. This becomes particularly challenging when prenatal phenotypes are non-specific, such as ventriculomegaly, fetal growth restriction, or oligohydramnios, which can result from a wide range of genetic and non-genetic causes. Therefore, non-targeted approaches such as ES and genome sequencing (GS) may be more suitable in such cases.

### 5.2. Exome Sequencing

ES is an increasingly utilized tool in prenatal diagnostics, particularly for fetuses with ultrasound anomalies and nondiagnostic karyotype and CMA results. ES targets the coding exons and exon–intron boundaries, typically covering up to ±20 base pairs into the intronic regions to capture potential splice-site variants. These cover approximately 1% to 2% of the genome, but contain most known disease-causing variants [81]. ES is typically performed using a capture-based enrichment method followed by high-throughput sequencing, achieving an average sequencing depth of approximately 100x to ensure reliable variant detection.

Several studies have evaluated the diagnostic yield of ES in prenatal settings, demonstrating wide variability based on fetal phenotype and selection criteria [9,82,83,84]. Higher yields have been reported in fetuses with multisystem anomalies or in specific organ system involvement, such as skeletal dysplasias [9,83], compared to those with isolated findings [82,84]. Two of the largest prospective studies, by Lord et al. [85] and Petrovski et al. [86], evaluated the clinical utility of trio-based prenatal ES in fetuses with structural anomalies detected by ultrasound and nondiagnostic karyotype and CMA, reporting diagnostic yields of 8.5% and 10.3%, respectively. Both studies underscore that yield varies by the type of anomaly. A systematic review and meta-analysis further supported these findings, with an overall diagnostic yield of 31%, increasing to 42% in cases with a suspected monogenic etiology, compared to 15% in unselected cases [9]. Consistent with these findings, the International Society for Prenatal Diagnosis (ISPD) recommends genome-wide (exome or genome) sequencing when karyotype and CMA are nondiagnostic, especially in fetuses with multiple anomalies or that have a suspected monogenic condition [87].

### 5.3. Genome Sequencing

GS extends beyond the exome to include most non-coding regions, offering a more comprehensive view of the genome. This broader scope allows for the detection of variants that may be missed by ES, including deep intronic variants, noncoding regulatory variants, and copy number changes. GS is typically performed at an average depth of 30–40x and does not rely on capture-based enrichment, reducing bias and improving uniformity across the genome. GS offers a comprehensive diagnostic approach by enabling the simultaneous detection of SNVs and CNVs at a high resolution, down to the exonic level of coding genes. Unlike traditional sequential testing workflows in which CMA is followed by ES if negative, GS can consolidate the capabilities of both into a single, streamlined assay, but this has yet to be fully validated on prenatal clinical samples. Beyond SNV and CNV detection, GS can also detect regions of homozygosity (ROH) and uniparental disomy (UPD) when parental samples are available, which is crucial for diagnosing recessive conditions and imprinting disorders. Furthermore, GS has the added advantage of detecting tandem repeat expansions and mitochondrial DNA abnormalities, both of which are typically missed by ES and CMA [88].

Prenatal GS for fetal anomalies has a comparable diagnostic yield to the combined performance of CMA and ES [89]. A recent study by Westenius et al. reported that clinical prenatal trio GS detected a causative variant in 26% (13/50) of fetuses with congenital malformations after nondiagnostic testing for CNVs and aneuploidy [90]. It is important to note that the analyses in these GS studies primarily focused on detecting SNVs and CNVs. As such, the reported diagnostic yield may underestimate the full potential of GS, since repeat expansions, mitochondrial DNA abnormalities, and structural variants were not systematically evaluated in these prenatal GS studies.

Although GS has clear technical advantages, its widespread adoption is currently limited by less mature bioinformatic pipelines, particularly for repeat detection due to the limitations of short-read sequencing. Additionally, the higher costs of sequencing, data storage, and interpretation remain significant challenges for widespread adoption in laboratories. As sequencing costs continue to decline, GS is expected to become more widely used in prenatal diagnostics and may eventually replace existing testing strategies, serving as a single comprehensive platform for detecting both chromosomal and single-gene disorders.

### 5.4. Considerations for Data Interpretation in Prenatal Sequencing

The interpretation of variants detected on prenatal ES/GS can be challenging because the prenatal-onset phenotypes of many disorders are poorly characterized and the phenotype information available from fetal imaging is often limited [91]. To address this gap, the Human Phenotype Ontology consortium has actively collaborated with an international network of fetal medicine experts to expand and refine ontology terms specific to prenatal development. This global initiative significantly enhances the capability to systematically capture, analyze, and interpret prenatal phenotypic data [92]. Additionally, broader data sharing through initiatives like the Fetal Sequencing Consortium is critical to improve our understanding of both causative and incidental findings in fetal ES and GS [92,93,94].

To improve diagnostic yield, trio-based sequencing is often preferred to singleton because it enables identification of de novo variants, determination of inheritance patterns, and improved interpretation of variant penetrance [9]. Trio analysis also significantly reduces the number of VUS identified by ES, decreasing from 15% in proband-only analysis to 6% with trio analysis. Most guidelines, including those of the ACMG, ISPD, and Canadian College of Medical Geneticists (CCMG), recommend reporting only VUS that are strongly associated with the fetal phenotype, typically after consensus is reached through multidisciplinary team review [75,95]. However, criteria and practice for reporting VUS can vary widely between different laboratories. ES/GS might generate incidental or secondary findings. Secondary findings are pathogenic or likely pathogenic variants in a defined list of medically actionable genes (e.g., those linked to hereditary cancer or cardiovascular disease), where early intervention can reduce morbidity and mortality [96]. Incidental findings also involve pathogenic variants unrelated to the testing indication, but include genes that fall outside the secondary findings list. They can involve genes associated with childhood conditions without a (distinct) prenatal phenotype as well as late-onset conditions such as neurological or neuromuscular disorders. As outlined in an ISPD position statement [87], these findings can raise important ethical considerations in the prenatal context. These highlight the importance of comprehensive genetic counseling both before and after testing to support informed decision-making and appropriate result disclosure.

Reanalysis of fetal ES and GS may be considered in cases with only VUS reported or with nondiagnostic or negative results, especially when new phenotypic features are observed after birth. However, it is often constrained by challenges related to cost, logistics, and the timing of reanalysis. A comparative overview of the major prenatal screening and diagnostic modalities is provided in Table 1.

## 6. Emerging Technologies

### 6.1. Optical Genome Mapping

Optical genome mapping (OGM) is a new cytogenetic tool that labels ultrahigh-molecular-weight (UHMW) DNA molecules and analyzes their pattern to detect copy number and structural variants across the genome [97]. OGM achieves average read lengths exceeding 200 Kb, allowing for detection of CNVs, balanced translocations, inversions, and complex chromosomal rearrangements as small as 500 bp. This technology has been clinically validated for detecting prenatal chromosomal aberrations with high sensitivity (99.5%) and specificity (100%), closely approximating that of conventional methods including karyotyping, CMA, and fluorescence in situ hybridization (FISH) [98]. OGM can also screen for fragile X syndrome (FXS) expansions using the EnFocus Fragile X analysis pipeline, which can detect full CGG repeat expansions, holding promise for future expanded screening. It has been proposed that by consolidating multiple cytogenomic analyses into a single assay, OGM could streamline prenatal diagnostic workflows [98], but this has not yet been validated in large studies. The requirement for UHMW DNA precludes using OGM on DNA extracted by commonly used protocols in clinical laboratories.

### 6.2. RNA Sequencing

RNA sequencing (RNA-seq) is also gaining traction as a functional prenatal diagnostic tool that can complement DNA sequencing to resolve VUS by detecting aberrant splicing, expression outliers, and monoallelic expression. It holds future potential for transcriptome-driven diagnosis, especially when DNA sequencing alone yields inconclusive results [99]. Such a reflex RNA-seq approach can clarify the functional impact of splicing variants, with approximately one-third of suspected cases demonstrating aberrant splicing [100,101], but translating it to the prenatal setting is challenging because of the added time required to obtain interpretable results and current limited knowledge about gene expression and splicing patterns in amniocytes or chorionic villi. Further development and validation before routine integration into prenatal diagnostics will be needed.

### 6.3. Long-Read Sequencing

Another emerging technology is long-read sequencing. Depending on the sequencing platform used, it can achieve read lengths ranging from 10–15 Kb to as high as 50–100 Kb, substantially longer than the standard short reads of a few hundred nucleotides [102]. This enables detection of complex structural variants, tandem repeat expansions, and methylation patterns at kilobase resolution. Long-read sequencing can also distinguish highly homologous genes from their pseudogenes and detect inversions, including those involving genes frequently included in carrier screening, such as *CYP21A2*, *SMN1/SMN2*, and *F8*, which present challenges for conventional sequencing methods [103,104,105,106]. In addition, long read technology enables the detection and characterization of long cfDNA fragments (>600 bp) and has potential to enhance resolution for fragmentomics [107]. Specifically, the analysis of long cfDNA fragments shows promise for early detection of preeclampsia through size-based classifiers and offers potential for assessing fetal inheritance patterns using methylation markers [108]. However, long cfDNA fragments make up less than 10% of total cfDNA in maternal plasma, while short fragments account for over 90%, which presents analytical challenges, but efforts to enrich these underrepresented fragments are ongoing [109]. An overview of these emerging technologies and their detectable genomic abnormalities is shown in Figure 2.

## 7. Future Methods

Despite developments in cfDNA analysis, cell-based noninvasive prenatal testing (cbNIPT) using intact fetal cells, primarily extravillous trophoblasts (EVTs), or fetal nucleated red blood cells (fNRBCs), isolated from maternal blood for genetic analysis is being actively investigated. The principal advantage of cbNIPT is its ability to analyze pure fetal DNA, free from maternal DNA admixture, which is not possible with cfDNA analysis [110,111,112]. Extravillous trophoblasts (EVTs) can also be retrieved from the cervix using Pap smear or cytobrush techniques and have been investigated for their diagnostic potential [113,114]. Immunoaffinity and microchip-based technologies, including antibody- and aptamer-mediated enrichment, are the leading methods for isolating trophoblasts and fNRBCs, with microfluidic platforms showing promise for clinical scalability [115,116]. However, challenges remain that should be addressed in future research, including the rarity of fetal cells in maternal circulation, the difficulty in recovering cells in all pregnancies, the complexity of isolation protocols, and variability in cell cycle stage or cell apoptosis that may affect genetic interpretation [110]. Overall, cbNIPT holds clinical potential for early, noninvasive, and comprehensive prenatal genetic diagnosis, but further technological refinement and validation are required before widespread clinical implementation [110,117].

## 8. Conclusions

Rapid developments in prenatal genomic technologies are reshaping the landscape of prenatal and reproductive testing and screening by enhancing the ability to identify a wide array of genetic conditions during pregnancy. Methods such as prenatal cfDNA-based testing and screening, CMA, and exome sequencing are now well established. Genome sequencing is increased and holds the promise of a streamlined and more comprehensive prenatal diagnostic alternative. Innovations like optical genome mapping, RNA-seq, and long-read sequencing promise to address remaining gaps in variant interpretation and detection of complex genomic abnormalities. Accessibility is influenced by the availability of testing platforms (e.g., CMA, exome/genome sequencing, cfDNA), local laboratory capabilities, and access to specialized counseling, with disparities persisting due to geography, insurance coverage, and institutional resources. The cost-effectiveness of testing varies by modality, indication, and patient population. Timely return of results is also critical. This varies by laboratory and type of assay, with cfDNA and targeted panels typically yielding results faster than CMA and exome/genome sequencing. As these tools advance toward broader clinical use, challenges remain in variant interpretation, counseling, equitable access, and ethical considerations. Discordance in VUS interpretation and potential reclassification can create uncertainty for clinical decision-making and may lead to parental anxiety, highlighting the importance of comprehensive counseling. Disparities in access, particularly in underrepresented populations, further compound these issues. Future integration into prenatal care of emerging technologies will require careful validation, guideline development, and expert bioinformatic support to translate their potential into routine prenatal care.

## Figures and Tables

**Figure 1 diagnostics-15-02579-f001:**
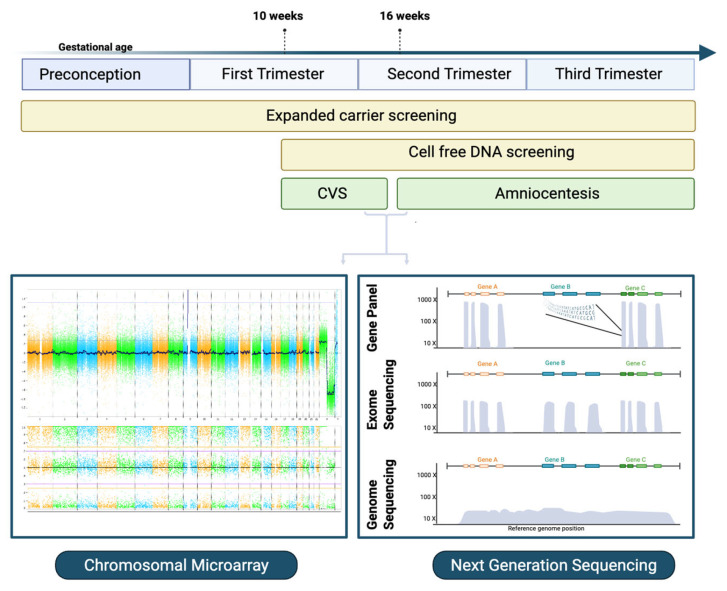
Timeline and current assays used for prenatal genetic testing. Schematic illustrating the timing and range of genetic testing options from preconception through pregnancy. Expanded carrier screening (yellow bar) is typically offered preconception or any time in pregnancy. Cell-free DNA (cfDNA) can be offered from 10 weeks’ gestation onwards, while diagnostic procedures such as chorionic villus sampling (CVS) and amniocentesis (green boxes) are available later. Examples of downstream laboratory methods include targeted gene panels, exome sequencing, and genome sequencing (left), as well as chromosomal microarray (CMA) analysis (right). CMA plot from deidentified patient tested through Baylor Genetics.

**Figure 2 diagnostics-15-02579-f002:**
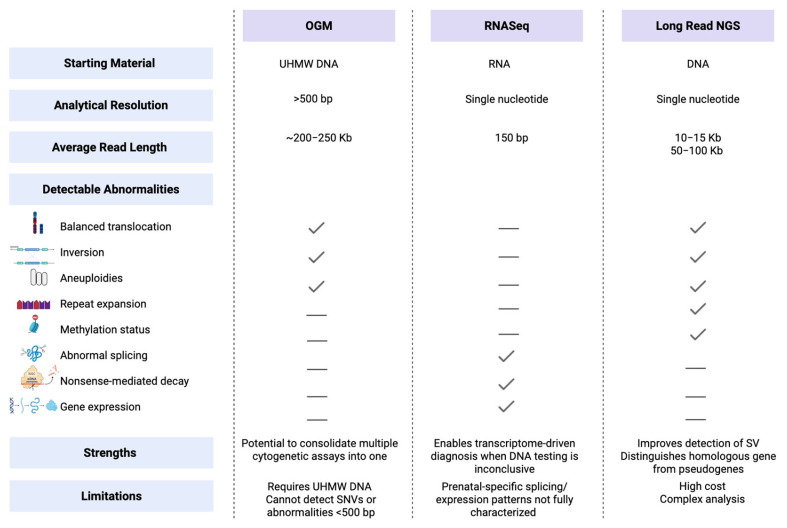
Emerging technologies and corresponding genomic abnormalities detectable in the prenatal setting. Comparison of three emerging genomic technologies—optical genome mapping (OGM), RNA sequencing (RNASeq), and long-read next-generation sequencing (NGS)—and their ability to detect various types of abnormalities. Bp, base pair; Kb, kilobase; SV, structural variants; UHMW DNA, ultrahigh-molecular-weight DNA; VUS, variants of uncertain significance.

**Table 1 diagnostics-15-02579-t001:** Comparative evidence of current prenatal genetic testing modalities.

Method		Detectable Abnormalities	Performance	Ref
Screening Tests	ECS	Targeted panel ○ Genes associated with AR/X-linked diseases ○ SNV & small indels ○ Deletion/duplications (selected genes)	○ DR varies by panel size, content, and indication	[2]
	cfDNA	Genome-wide approach ○ Aneuploidies ○ Microdeletion/microduplication	○ T21: Se 98.8%, Sp 99.96%, PPV 91.8%, NPV 100% ○ T18: Se 98.8%, Sp 99.9%, PPV 65.8%, NPV 100% ○ T13: Se 100%, Sp 99.96%, PPV 37.2%, NPV 100% ○ SCAs: DR 99.6%, Sp 99.8%, PPVs vary ○ 22q11.2DS: PPV 52.6%, FPR 0.05% ○ RATs: pooled PPV 11.46%	[3,5,44,45]
	cfDNA—SGD	Targeted panel ○ Genes with high incidence of de novo variants ○ SNVs and small indels	○ Se >99%, Sp >99%, PPV >99%, NPV >99% ○ DR varies by gene	[6,53]
Diagnostic Tests	CMA	Genome-wide approach ○ Aneuploidies ○ CNV ○ ROH	○ Diagnostic yield: 6.0% (fetuses with anomalies), 1.7% (AMA)	[7,71]
	LP-GS	Genome-wide approach ○ Aneuploidies ○ CNV ○ ROH	○ Diagnostic yield 8%–23%	[8,76,77]
	Exome Sequencing	Exome-wide approach ○ SNVs and small indels	○ Diagnostic yield 15%–42% (depending on phenotype)	[9]
	Genome Sequencing	Genome-wide approach ○ Aneuploidies ○ CNV ○ SNVs and small indels ○ Repeat expansions	○ Diagnostic yield 26% (fetuses with anomalies)	[90]

Key prenatal genetic screening and diagnostic technologies. For each modality, the detectable abnormalities are outlined, along with representative performance metrics, diagnostic yield, and detection rates. References correspond to representative studies supporting the summarized evidence. Abbreviations: AMA, advanced maternal age; AR, autosomal recessive; bp, base pair; cfDNA, cell-free DNA; CMA, chromosomal microarray; CNV, copy number variant; DR, diagnostic rate; DS, deletion syndrome; ECS, expanded carrier screening; FPR, false-positive rate; kb, kilobase; LP-GS, low-pass genome sequencing; NGS, next-generation sequencing; NPV, negative predictive value; OGM, optical genome mapping; PPV, positive predictive value; RAT, rare autosomal trisomy; ROH, regions of homozygosity; SCA, sex chromosome aneuploidy; Se, sensitivity; SGD, single-gene disorder; SNP, single-nucleotide polymorphism; SNV, single-nucleotide variant; Sp, specificity; SV, structural variant; T, trisomy; UPD, uniparental disomy; VUS, variants of uncertain significance.

## Data Availability

No new data were created or analyzed in this study. Data sharing is not applicable to this article.

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
