# Peer review of "Advancements in Prenatal Genetic Screening and Testing: Emerging Technologies and Evolving Applications"

_diagnostics, 2025, doi:10.3390/diagnostics15202579_

Round 1
Reviewer 1 Report
Comments and Suggestions for Authors
Manuscript 'Advancements in Prenatal Genetics: Emerging Technologies and Evolving Applications' presents thorough overview of current state of the art in the field of prenatal genetic diagnostics.
The review is clear, comprehensive and relevant to the field. The cited references are mostly recent publications (within the last 5 years) and relevant. The statements and conclusions drawn are coherent and supported by the listed citations. The tables and images are appropriate and they properly show the data. They are easy to interpret and understand.
I recommend this manuscript to be accepted after minor corrections are done. In my opinion, chapter 'Cell-based Noninvasive Prenatal Testing' should be further expanded with more details regarding technical challenges in implementing analysis of fetal cells in routine practice.
Author Response
Reviewer 1: Comments and Suggestions for Authors
Manuscript “Advancements in Prenatal Genetics: Emerging Technologies and Evolving Applications” presents thorough overview of current state of the art in the field of prenatal genetic diagnostics.
Comment 1: The review is clear, comprehensive and relevant to the field. The cited references are mostly recent publications (within the last 5 years) and relevant. The statements and conclusions drawn are coherent and supported by the listed citations. The tables and images are appropriate, and they properly show the data. They are easy to interpret and understand.
Response 1: We thank the reviewer for the positive feedback and are pleased that the review was found to be clear, comprehensive, and well-supported.
Comment 2: I recommend this manuscript to be accepted after minor corrections are done. In my opinion, chapter 'Cell-based Noninvasive Prenatal Testing' should be further expanded with more details regarding technical challenges in implementing analysis of fetal cells in routine practice.
Response 2: We thank the reviewer for this helpful suggestion. In response, we have expanded the section on cell-based NIPT (starting at line 515) to include additional details on technical challenges. The revised text now states: “However, challenges remain that should be addressed in future research, including the rarity of fetal cells in maternal circulation, the difficulty with recovering cells in all pregnancies, the complexity of isolation protocols, variability in cell cycle stage or cell apoptosis that may affect genetic interpretation.”
Reviewer 2 Report
Comments and Suggestions for Authors
This is an exceptionally comprehensive and timely review that provides a clear and valuable overview of the rapidly evolving landscape of prenatal genetic testing. The manuscript's major strength lies in its logical organization and thorough coverage, seamlessly guiding the reader from established screening modalities like carrier screening and cfDNA analysis to advanced diagnostic tools such as exome/genome sequencing and finally to emerging technologies. The consistent emphasis on clinical utility, limitations, and professional guideline recommendations makes it an immensely practical resource for clinicians and laboratorians alike. The scope is impressive, effectively balancing technical detail with clinical relevance.
To further enhance the manuscript's impact, minor refinements are suggested. The figures, particularly the timeline in Figure 1, could be augmented with specific gestational ages and a clearer visual distinction between screening and diagnostic tests. Figure 2 could be transformed into a more intuitive heat map to instantly convey the capabilities of each emerging technology. Additionally, while alluded to, the profound challenges associated with variants of uncertain significance (VUS), the complexities of genetic counseling, and critical issues of equitable access could be underscored more prominently in the conclusion to fully highlight the remaining barriers to implementation. A final careful proofread would ensure consistency in terminology and formatting. These small adjustments would solidify an already outstanding and informative review.
Author Response
Reviewer 2: Comments and Suggestions for Authors
Comment 1: This is an exceptionally comprehensive and timely review that provides a clear and valuable overview of the rapidly evolving landscape of prenatal genetic testing. The manuscript's major strength lies in its logical organization and thorough coverage, seamlessly guiding the reader from established screening modalities like carrier screening and cfDNA analysis to advanced diagnostic tools such as exome/genome sequencing and finally to emerging technologies. The consistent emphasis on clinical utility, limitations, and professional guideline recommendations makes it an immensely practical resource for clinicians and laboratorians alike. The scope is impressive, effectively balancing technical detail with clinical relevance.
Response 1: We sincerely thank the reviewer for the very positive and encouraging feedback. We are pleased that the review’s organization, scope, and clinical emphasis were found to be valuable and relevant.
Comment 2: To further enhance the manuscript's impact, minor refinements are suggested. The figures, particularly the timeline in Figure 1, could be augmented with specific gestational ages and a clearer visual distinction between screening and diagnostic tests. Figure 2 could be transformed into a more intuitive heat map to instantly convey the capabilities of each emerging technology. Additionally, while alluded to, the profound challenges associated with variants of uncertain significance (VUS), the complexities of genetic counseling, and critical issues of equitable access could be underscored more prominently in the conclusion to fully highlight the remaining barriers to implementation. A final careful proofread would ensure consistency in terminology and formatting. These small adjustments would solidify an already outstanding and informative review.
Response 2: We thank the reviewer for these constructive suggestions and made the following changes:
a) We revised Figure 1 to include specific gestational ages and to clarify the distinction between screening and diagnostic tests.
b) While limited by the number of figures allowed, we modified Figure 2 to include additional details on emerging technologies.
c) We also expanded the conclusion (line 539) to more explicitly address challenges related to VUS interpretation, counseling, equitable access, and ethical considerations: “As these tools advance toward broader clinical use, challenges remain in variant interpretation, counseling, equitable access, and ethical considerations. Discordance in VUS interpretation and potential reclassification can create uncertainty for clinical decision-making and may lead to parental anxiety, highlighting the importance of comprehensive counseling. Disparities in access, particularly in underrepresented populations, further compound these issues. Future integration into prenatal care of emerging technologies will require careful validation, guideline development, and expert bioinformatics support to translate their potential into routine prenatal care.”
d) We carefully proofread the manuscript to ensure consistency in terminology and formatting.
Reviewer 3 Report
Comments and Suggestions for Authors
- The narrative review “Advancements in Prenatal Genetics: Emerging Technologies and Evolving Applications’’ is informative, discussing key advances in prenatal genetic testing and providing an overview of different diagnostic tests, including expanded carrier sequencing (ECS), cell-free DNA (cfDNA) testing, Chromosomal microarray analysis, and next-generation sequencing. A few comments are listed below to further revise it.
- The title could be “Advancements in Prenatal Genetic Screening: Emerging Technologies and Evolving Applications”
- The methodology used in framing this narrative review is missing.
- The review could be enhanced by including a brief overview of commonly used prenatal biochemical tests (β-hCG, AFP, PAPP-A, uE3, and inhibin-A) performed before genetic testing.
- Details on what type of biological sample is needed for the test are missing.
- Including practical considerations like test accessibility, cost-effectiveness, and turnaround time for result reporting could benefit the review.
- Including a comparative table with quantitative data would be preferable to providing the specific outcomes of each method.
- Line 492- Grammatical error- sequencing is increased
- A comparative summary of all the discussed tests would enable readers to understand the strengths and limitations of each test easily.
- Discussion on emerging technologies, such as methylation-based tests and CRISPR-based platforms, is lacking.
- Figure 2 can be enhanced by converting it to a table and incorporating additional information, such as the ideal quantitative ranges for each assay and the strengths and weaknesses of each test.
- Bold and highlighted sections (Lin 56, Line 123-25) can be avoided in a review.
see before
Author Response
Reviewer 3: Comments and Suggestions for Authors
Comment 1: The narrative review “Advancements in Prenatal Genetics: Emerging Technologies and Evolving Applications’’ is informative, discussing key advances in prenatal genetic testing and providing an overview of different diagnostic tests, including expanded carrier sequencing (ECS), cell-free DNA (cfDNA) testing, Chromosomal microarray analysis, and next-generation sequencing. A few comments are listed below to further revise it.
Response 1: We thank the reviewer for recognizing the scope and informativeness of our review. We look forward to addressing the specific comments provided below to further strengthen the manuscript.
Comment 2: The title could be “Advancements in Prenatal Genetic Screening: Emerging Technologies and Evolving Applications”
Response 2: We thank the reviewer for this suggestion. To provide additional clarity, we have modified the title to read: “Advancements in Prenatal Genetic Screening and Testing: Emerging Technologies and Evolving Applications”. However, our review covers a broad range of genetic assays and their applications in prenatal settings, extending beyond screening alone. We therefore believe the slightly revised title now more accurately reflects the scope of the manuscript. As this was an invited review, and neither the section editors nor the other reviewers raised concerns about the title, we elected to keep it as is. That said, we are happy to work with the editors if they feel a change is needed.
Comment 3: The methodology used in framing this narrative review is missing.
Response 3: We thank the reviewer for this comment. We have now added a Materials and Methodology section (beginning at line 66 of the revised manuscript) to clarify the narrative review approach, including the literature search strategy and selection criteria. The section reads as follows: "This narrative review was informed by a targeted literature search in PubMed and Google Scholar using key terms such as ‘prenatal genetics,’ ‘carrier screening,’ ‘cell-free DNA,’ ‘chromosomal microarray,’ and ‘exome sequencing.’ Priority was given to publications from the last five years, supplemented by landmark earlier studies and current clinical guidelines. Articles were selected for relevance to prenatal applications, clinical utility, and technological advances. The review was not intended as a systematic review but rather as an overview of key developments and evolving applications in the field."
Comment 4: The review could be enhanced by including a brief overview of commonly used prenatal biochemical tests (β-hCG, AFP, PAPP-A, uE3, and inhibin-A) performed before genetic testing.
Response 4: We thank the reviewer for this suggestion. As the focus of our review is on genetic testing technologies, we consider a detailed overview of biochemical screening tests to be beyond the intended scope. However, we added the following clarification at line 152: “Prior to implementation of cfDNA-based aneuploidy screening, biochemical serum screening (β-hCG, AFP, PAPP-A, uE3, inhibin-A), often combined with nuchal translucency, formed the basis of combined, triple, and quadruple screens for aneuploidy risk, particularly trisomies 21 and 18, and remains in use in some settings.”
Comment 5: Details on what type of biological sample is needed for the test are missing.
Response 5: We thank the reviewer for this comment. While the type of biological sample is alluded to in Figure 1, we have now clarified this further by adding the following text at lines 39–46 of the manuscript: “Expanded carrier screening (ECS) enables pan-ethnic detection of a broad range of autosomal recessive and X-linked conditions in prospective parents and is performed on a maternal blood sample [2]. Cell-free DNA (cfDNA) screening, also based on a maternal blood sample, noninvasively assesses fetal risk for common aneuploidies, and with advancing technology, has expanded to include sex chromosome aneuploidy (SCA) [3], rare autosomal trisomies (RATs) [4], both common and rare copy number variants (CNVs) [5], and single-gene disorders [6]. Among diagnostic tests performed prenatally, usually done on samples obtained through chorionic villus sampling (CVS) or amniocentesis, chromosomal microarray analysis (CMA), which detects aneuploidy and CNVs, including submicroscopic CNVs not visible through conventional karyotyping, is the recommended diagnostic test for pregnancies complicated by fetal anomalies [7].”
Comment 6: Including practical considerations like test accessibility, cost-effectiveness, and turnaround time for result reporting could benefit the review.
Response 6: We thank the reviewer for this helpful suggestion. In response, we have added the following to the conclusion (starting at line 533): “Accessibility is influenced by the availability of testing platforms (e.g., CMA, exome/genome sequencing, cfDNA), local laboratory capabilities, and access to specialized counseling, with disparities persisting due to geography, insurance coverage, and institutional resources. The cost-effectiveness of testing varies by modality, indication, and patient population. Timely return of results is also critical, with differences that vary by laboratory and type of assay, with cfDNA and targeted panels typically yielding results faster than CMA and exome/genome sequencing.”
Comment 7: Including a comparative table with quantitative data would be preferable to providing the specific outcomes of each method.
Response 7: We thank the reviewer for this suggestion. While we appreciate the value of a comparative table, we have addressed the outcomes of each genetic test within the text, and Figure 2 illustrates the different outcomes of the emerging technologies. We believe this effectively conveys the distinctions and key points. We also modified Figure 2.
Comment 8: Line 492- Grammatical error- sequencing is increased
Response 8: We thank the reviewer for noting this error. The sentence at line 492 has been corrected accordingly.
Comment 9: A comparative summary of all the discussed tests would enable readers to understand the strengths and limitations of each test easily.
Response 9: We thank the reviewer for this excellent suggestion. We agree that a comparative summary would be useful; however, due to figure limitations and to avoid redundancy with content already presented in the text and figures, we did not include an additional table. However Figure 2 was revised as outlined below (comment 11).
Comment 10: Discussion on emerging technologies, such as methylation-based tests and CRISPR-based platforms, is lacking.
Response 10: We thank the reviewer for this comment. While CRISPR-based platforms are important, they are not currently applied as emerging prenatal diagnostic technologies. However, to address the comment on methylation assays, we added the following to line 53: “Methylation assays are also clinically available for prenatal diagnosis of imprinting disorders, though a detailed discussion is beyond the scope of this review.” We also added the reference PMID: 37340544.
Comment 11: Figure 2 can be enhanced by converting it to a table and incorporating additional information, such as the ideal quantitative ranges for each assay and the strengths and weaknesses of each test.
Response 11: We thank the reviewer for this suggestion. The strengths, weaknesses, and relevant considerations for each assay are addressed in the text, and we have added more information to Figure 2 as additional rows. Since some of these tests are not yet clinically available, including detailed strengths and weaknesses in the figure may be premature, but we do provide this context in the text (Section 7).
Comment 12: Bold and highlighted sections (Line 56, Line 123-25) can be avoided in a review.
Response 12: We thank the reviewer for this helpful suggestion. The bold formatting at lines 56 and 123–125 has been removed as recommended.
Round 2
Reviewer 3 Report
Comments and Suggestions for Authors
If it looks critically, neither of the figures is synthesised from actual data. Figure 2 lacks novel information as it is already known. Therefore, the author must include one data Table that consists of cited evidence using detectable abnormalities in the subject matter proposed in this review. There are 116 cited references are included, yet it is lacking.
Including a table with quantitative data would be preferable to providing the specific outcomes of each method. At present, the narrative review lacks actual ownership of its proposed view without such evidence.
Author Response
Reviewer 3
Comment 1: If it looks critically, neither of the figures is synthesised from actual data. Figure 2 lacks novel information as it is already known. Therefore, the author must include one data Table that consists of cited evidence using detectable abnormalities in the subject matter proposed in this review. There are 116 cited references are included, yet it is lacking.
Including a table with quantitative data would be preferable to providing the specific outcomes of each method. At present, the narrative review lacks actual ownership of its proposed view without such evidence.
Response 1: We thank the reviewer for this helpful suggestion, which has greatly improved our manuscript. Following this advice, we created a new Table 1 that compiles and summarizes the cited evidence, presenting quantitative data on detectable abnormalities across the discussed methods. We also further modified Figure 2 to complement this table and to better highlight the comparative aspects of the technologies. We believe these additions provide the critical data synthesis the reviewer recommended and strengthen the manuscript by giving readers a clearer, evidence-based understanding of the outcomes of each approach.
Additionally, we revised Section 5 (line 329) to include the following highlighted text:
“As more laboratories adopt NGS, low-pass genome sequencing (LP-GS) has emerged as a promising alternative, offering flexible CNV detection with reported coverage in prenatal studies ranging from 0.25× to 5× [8, 74–76]. Prenatal studies comparing LP-GS with CMA have reported diagnostic yields ranging from 8% to 23% [PMID: 37012053; PMID: 32451733]. Although the fundamental principles and clinical utility of LP-GS have been established, broader adoption may require further validation and endorsement by professional guidelines. In the long run, LP-GS holds potential as a more cost-effective and efficient diagnostic tool, particularly when integrated into a comprehensive sequencing strategy.”
Round 3
Reviewer 3 Report
Comments and Suggestions for Authors
The revised version has improved considerably. Few minor edits :
Line 66: Remove heading material and methods and integrate into the running text
Segregate emerging and future Technologies, possibly with a separate heading.
Author Response
Comment 1: Line 66: Remove heading “Material and Methods” and integrate into the running text.
Segregate emerging and future Technologies, possibly with a separate heading.
Response 1: We thank the reviewer for these helpful suggestions. We have removed the “Material and Methods” heading at line 66 and integrated the content into the running text. In addition, we have segregated the sections on emerging and future technologies and added a separate heading, as recommended.